# B-Cell Activation Biomarkers in Salivary Glands Are Related to Lymphomagenesis in Primary Sjögren’s Disease: A Pilot Monocentric Exploratory Study

**DOI:** 10.3390/ijms25063259

**Published:** 2024-03-13

**Authors:** Dario Bruno, Barbara Tolusso, Gianmarco Lugli, Clara Di Mario, Luca Petricca, Simone Perniola, Laura Bui, Roberta Benvenuto, Gianfranco Ferraccioli, Stefano Alivernini, Elisa Gremese

**Affiliations:** 1Clinical Immunology Unit, Fondazione Policlinico Universitario A. Gemelli-IRCCS, 00168 Rome, Italy; dariobrunomd@gmail.com (D.B.); simone.perniola@guest.policlinicogemelli.it (S.P.); 2Department of Medicine, University of Verona, 37129 Verona, Italy; 3Immunology Core Facility, Gemelli Science and Technology Park, Fondazione Policlinico Universitario A. Gemelli-IRCCS, 00168 Rome, Italy; barbara.tolusso@policlinicogemelli.it (B.T.); dimarioclara@gmail.com (C.D.M.); 4Rare Disease Unit, Meyer Children’s Hospital IRCCS, 50139 Florence, Italy; gianmarco.lugli@meyer.it; 5Rheumatology Division, Fondazione Policlinico Universitario A. Gemelli-IRCCS, 00168 Rome, Italy; luca.petricca@policlinicogemelli.it; 6Institute of Pathology, Fondazione Policlinico Universitario A. Gemelli-IRCCS, 00168 Rome, Italy; laura.bui@policlinicogemelli.it (L.B.); roberta.benvenuto@policlinicogemelli.it (R.B.); 7Department of Internal Medicine, Catholic University of the Sacred Heart, 00168 Rome, Italy; gianfranco.ferraccioli@unicatt.it

**Keywords:** primary Sjögren’s disease (primary Sjögren’s disease), lymphomagenesis, microRNA-155, BAFF-receptor, IL-6 receptor, biomarkers, epigenetic

## Abstract

Primary Sjögren’s disease is primarily driven by B-cell activation and is associated with a high risk of developing non-Hodgkin’s lymphoma (NHL). Over the last few decades, microRNA-155 (miR-155) has arisen as a key regulator of B-cells. Nevertheless, its role in primary Sjögren’s disease remains elusive. Thus, the purpose of this study was (i) to explore miR-155, B-cell activating factor (BAFF)-receptor (BAFF-R), and Interleukin 6 receptor (IL-6R) expression in the labial salivary glands (LSG) of patients with primary Sjögren’s disease, aiming to identify potential B-cell activation biomarkers related to NHL development. Twenty-four patients with primary Sjögren’s disease, and with available tissue blocks from a LSG biopsy performed at diagnosis, were enrolled. Among them, five patients developed B-cell NHL during follow-up (7.3 ± 3.1 years). A comparison group of 20 individuals with sicca disease was included. Clinical and laboratory parameters were recorded and the LSG biopsies were evaluated to assess local inflammation in terms of miR-155/BAFF-R and IL-6R expression. Stratifying the primary Sjögren’s disease cohort according to lymphomagenesis, miR-155 was upregulated in primary Sjögren’s disease patients who experienced NHL, more so than those who did not experience NHL. Moreover, miR-155 expression correlated with the focus score (FS), as well as BAFF-R and IL-6R expression, which were increased in primary Sjögren’s disease patients and in turn related to neoplastic evolution. In conclusion, epigenetic modulation may play a crucial role in the aberrant activation of B-cells in primary Sjögren’s disease, profoundly impacting the risk of NHL development.

## 1. Introduction

Primary Sjögren’s disease is a systemic autoimmune disease characterised by chronic inflammation of exocrine glands in the absence of other systemic connective tissue diseases. Conversely, secondary Sjögren’s disease is diagnosed when symptoms of Sjögren’s disease coexist with other systemic connective tissue diseases [1]. The prevalence of primary Sjögren’s disease is notably high, being nearly half as common as that of Rheumatoid Arthritis (RA) [2]. Dry mouth and dry eyes, due to lymphocytes’ infiltration of the salivary and lacrimal glands, are the clinical hallmarks, although a wide spectrum of extra-glandular manifestations (EGM) can occur [3]. Since almost every organ can potentially be involved [4], Sjogren’s disease is a term often used to describe the many ways in which a patient can be affected by this condition. The high risk of developing B-cell non-Hodgkin’s lymphoma (NHL) represents the major adverse outcome of the disease, occurring in 2–5% of patients with primary Sjögren’s disease [5].

The pathogenesis of primary Sjögren’s disease is the result of a complex network between the aberrantly activated immune system and epithelial cells, leading to dysfunction of exocrine glands. Several studies have clarified some pathogenetic mechanisms, highlighting the prominent role of plasmacytoid dendritic cells (pDC) in producing activated type-I interferon (IFN) in the earliest phase of the disease [3,6]. Breakdown of immunological tolerance is a common feature among models of primary Sjögren’s disease pathogenesis, underpinned by antigen-driven T cell-mediated polyclonal B-cell hyperfunction, resulting in the appearance of specific serum autoantibodies (anti-SSA and SSB) able to induce tissue inflammation and damage, the formation of tertiary lymphoid structures (TLSs) [6,7] in the salivary glands, and B-cell lymphomagenesis [5]. Novel insights have been acquired regarding genetic and epigenetic pathways supporting and perpetuating—together with environmental factors—chronic inflammation in primary Sjögren’s disease patients [8]. Concerning epigenetic mechanisms, it is becoming progressively evident that their ability to regulate gene expression may play a key role in autoimmune disorders [9,10,11,12,13]. One model of epigenetic control is exerted by microRNAs (miRNAs), which are small noncoding RNAs capable of binding to the 3′ untranslated region (UTR) of specific messenger RNAs (mRNAs), leading to repression of post-transcriptional genes. It is well-established that miRNAs have a crucial role in the regulation of multiple molecular pathways involved in the development, differentiation, and apoptosis of immune cells, including B-cells [9,10,11,12,13]. Consequently, a variety of studies have been conducted in an effort to investigate the aberrant expression of miRNAs in rheumatic diseases as well as in cancer in humans, paving the way for new possible therapeutic targets [14].

Over the last few decades, microRNA-155 (miR-155) has emerged as a key proinflammatory regulator in clinical and experimental arthritis and in systemic lupus erythematosus (SLE), targeting specific mRNAs, such as PU.1, SOCS-1, TAB-2, and c-Maf [9,10,13]. Moreover, it was demonstrated that miR-155 exerts its detrimental actions as an oncomiR in B-cell lymphoproliferative disorders, resulting in overexpression and correlating with a poor prognosis for Hodgkin’s lymphoma, primary mediastinal and diffuse large B-cell lymphomas, and paediatric Burkitt lymphoma [14]. Although the amount of data on miR-155 in primary Sjögren’s disease is still fairly poor, several lines of evidence suggest its important role in the development and maintenance of the chronic inflammatory process of the disease [8,13].

Particularly, in B-cells, miR-155 is inducible by several factors including B-cell activating factor (BAFF) and IL-6 [9,10]; BAFF is a member of the tumour necrosis factor (TNF) ligand family, known to be a key regulator of B-cell homeostasis, tolerance, and malignancy. Moreover, the BAFF-receptor (BAFF-R), which binds exclusively to BAFF, is pivotal in the early stages of B-cell maturation. Although the role of the BAFF pathway (BAFF/BAFF-R) has been deeply investigated in relation to several autoimmune diseases, such as rheumatoid arthritis (RA) [15] and systemic lupus erythematosus [16,17], data on primary Sjögren’s disease are mostly limited to soluble BAFF (sBAFF), and the possible correlation between its expression and lymphomagenesis in primary Sjögren’s disease remains to be elucidated. Regarding the IL-6/IL-6R axis, in a recent double-blind randomised placebo-controlled trial, tocilizumab failed to demonstrate effectiveness in 55 patients with primary Sjögren’s disease and moderate or high systemic disease activity [18]. However, the immunological impact of IL-6 remains crucial in primary Sjögren’s disease, considering its pivotal role in B-cell activation and T cell polarization along with BAFF and IL-21 [19,20].

Based on these issues, the aims of the present study were (i) to explore the expression of miR-155, BAFF-R, and IL-6-R in labial salivary gland (LSG) samples from primary Sjögren’s disease patients and evaluate their correlation with disease burden at baseline, and (ii) to examine their putative association with NHL development during follow-up.

## 2. Results

### 2.1. Baseline Demographic and Clinical Characteristics of Primary Sjögren’s Disease and Sicca Disease Cohorts

Table 1 summarizes the demographic, immunological, and clinical characteristics of the enrolled cohorts. Among the primary Sjögren’s disease patients, 21 were female (87.5%). At the time of diagnosis, the median age of the Sjögren’s disease cohort was 54.5 (47.3–64.8) years. In particular, primary Sjögren’s disease patients who developed NHL showed a higher median age at the time of their primary Sjögren’s disease diagnosis than primary Sjögren’s disease patients who did not develop NHL (*p* = 0.04) (Table 1). The median follow-up time from primary Sjögren’s disease to NHL diagnosis was 7.3 ± 3.1 years. All primary Sjögren’s disease patients who did not develop NHL had a follow-up time of at least 10 years (12.3 ± 1.4 years). Considering the immunological status, neither autoantibody positivity (IgA/IgM-RF, anti-Ro (SSA) and anti-La (SSB)) nor serum complement levels at diagnosis were significantly associated with the development of NHL in the primary Sjögren’s disease cohort (Table 1).

Nevertheless, the primary Sjögren’s disease patients who developed NHL had significantly higher IgA plasma levels (380.6 ± 73.9 mg/dL) at diagnosis than both the patients who did not experience NHL (188.9 ± 20.1 mg/dL; *p* = 0.002) and the sicca disease cohort (193.9 ± 20.0 mg/dL; *p* = 0.002) (Table 1). Conversely, primary Sjögren’s disease patients who developed NHL showed lower IgM plasma levels (69.4 ± 16.7 mg/dL) than patients without neoplastic transformation (132.6 ± 12.8 mg/dL; *p* = 0.03) and sicca disease patients (138.2 ± 12.6 mg/dL; *p* = 0.02). Furthermore, patients with primary Sjögren’s disease suffering from lymphomagenesis showed increased hypergammaglobulinemia (21.4 ± 1.8%) compared to patients who did not experience NHL evolution (15.9 ± 0.8; *p* = 0.006), as well as those with sicca syndrome (14.3 ± 0.6; *p* = 0.0001) (Table 1).

Regarding inflammatory markers, erythrocyte sedimentation rate (ESR) and C-reactive protein (CRP) plasma levels were comparable at diagnosis between the study cohorts. Finally, when stratifying the primary Sjögren’s disease cohort based on ESSDAI, no significant differences were observed among the different groups in terms of the chance of experiencing NHL development (Table 1).

### 2.2. Primary Sjögren’s Disease and Sicca Disease Patients Showed Different Histological Features in Terms of Labial Salivary Glands’ CD20^pos^ and CD138^pos^ Cell Enrichment

Each enrolled patient underwent LSG biopsy. As shown in Table 1, primary Sjögren’s disease patients showed significantly higher values of FS than sicca disease controls (3.5 ± 0.3 vs. 0.0 ± 0.0; *p* < 0.0001). Nevertheless, when stratifying the primary Sjögren’s disease cohort based on lymphomagenesis, there were no significant differences in terms of FS between patients who developed NHL (3.2 ± 0.6) and those who did not experience neoplastic transformation (3.6 ± 0.4; *p* = ns) (Table 1).

Immunohistochemical analysis (Figure 1A,C) revealed that primary Sjögren’s disease patients were characterized by higher IHC scores for CD20^pos^ cells (1.67 ± 0.60) than sicca disease patients (0.62 ± 0.45; *p* < 0.0001) (Figure 1B), regardless of NHL development (1.63 ± 0.64 in primary Sjögren’s disease not developing NHL vs. 1.80 ± 0.45 in primary Sjögren’s disease developing NHL; *p* = ns).

Regarding plasma cell evaluation, CD138^pos^ cells’ IHC score was significantly higher in LSG samples from primary Sjögren’s disease patients (1.82 ± 0.59) compared to those of sicca disease patients (1.19 ± 0.63; *p* = 0.001), directly correlating with CD20 IHC scores in patients suffering from primary Sjögren’s disease (R = 0.662; 95% confidence interval (95% CI) 0.451–0.808; *p* < 0.0001) (Figure 1D,E). Similarly, IHC analysis showed that primary Sjögren’s disease patients had comparable levels of CD138^pos^ cells regardless of lymphomagenesis (1.75 ± 0.64 primary Sjögren’s disease not developing NHL vs. 2.06 ± 0.72 primary Sjögren’s disease developing NHL; *p* = ns).

### 2.3. miR-155 Is Overexpressed in the Labial Salivary Glands of Patients with Primary Sjögren’s Disease, Being Related to Lymphomagenesis and Degree of Focal Inflammation

qPCR of miR-155 expression was performed on LSG samples from all study cohorts.

As shown in Figure 2A, the minor salivary glands of primary Sjögren’s disease patients showed an increased expression of miR-155 (fold change: 6.44 ± 1.62) compared to the minor salivary glands of patients with sicca disease (fold change: 1.44 ± 0.27; *p* = 0.001). Remarkably, when stratifying primary Sjögren’s disease patients based on NHL development during the follow-up, miR-155 expression was significantly higher in patients who developed NHL (fold change: 12.12 ± 3.18) compared to both primary Sjögren’s disease patients who did not experience NHL transformation (fold change: 4.94 ± 1.74; *p* = 0.037) and the sicca disease cohort (fold change: 1.44 ± 0.27; *p* = 0.001) (Figure 2B).

Interestingly, the receiver operating characteristic (ROC) curve analysis revealed that a 3.65-fold increase of miR-155 expression in LSGs had significant capacity to distinguish patients with primary Sjögren’s disease (50.0%) from those with sicca syndrome (5%; *p* = 0.001) (area under the curve [AUC] 0.780 [95% confidence interval (95% CI) 0.645–0.915] [*p* < 0.0001]; sensitivity: 50%; specificity: 99.5%) (Figure 2C and Appendix A).

Moreover, considering the primary Sjögren’s disease cohort, patients having a ≥5.0-fold increase of miR-155 expression in their LSGs were more likely to experience NHL transformation compared to patients who exhibited a fold increase of <5.0 (100% of patients who developed NHL vs. 15.8% of patients who did not develop NHL, *p* < 0.0001) (Area Under the Curve [AUC] 0.884 [95% confidence interval (95% CI) 0.744–1.024] [*p* < 0.0001]; sensitivity 100%; specificity 84.2%) (Figure 2D and Appendix A).

Additionally, LSG miR-155 expression was found to slightly correlate with FS (R = 0.355; *p* = 0.09) in the primary Sjögren’s disease cohort (Figure 2E). Moreover, when stratifying the primary Sjögren’s disease cohort according to FS, patients with an FS between 5 and 7 (maximum value in the study population) exhibited significantly higher expressions of miR-155 than those who had an FS between 1 and 4 (*p* = 0.0025) (Figure 2F).

Finally, miR-155 expression in LSGs from primary Sjögren’s disease patients directly correlated with hypergammaglobulinemia (R = 0.662; *p* = 0.001), supporting its prominent role in B-cell activation (Figure 3A).

### 2.4. BAFF-R and IL-6R Expression Are Upregulated in Labial Salivary Glands of Patients with Primary Sjögren’s Disease, Being Related to NHL Development and Directly Correlating with miR-155 Levels

Since B-cell activation is deemed to be pivotal in primary Sjögren’s disease pathogenesis as well as in NHL development, the expression of BAFF-R and IL-6R in the LSGs of primary Sjögren’s disease and sicca disease patients was explored. The salivary glands of primary Sjögren’s disease patients who experienced neoplastic progression were enriched with BAFF-R (fold change: 4.78 ± 1.61) compared to both primary Sjögren’s disease patients who did not develop NHL (fold change: 1.05 ± 0.20; *p* = 0.002) and sicca disease patients (fold change: 0.52 ± 0.06; *p* = 0.003) (Figure 3B).

Regarding IL-6R expression, the minor salivary glands of primary Sjögren’s disease patients showed a 7.36 ± 2.43-fold increase compared to patients with sicca disease (fold change: 1.81 ± 0.44; *p* = 0.005). In particular, when stratifying primary Sjögren’s disease patients based on neoplastic evolution during the follow-up, IL-6R expression was significantly higher in patients who developed NHL (fold change: 21.20 ± 9.62) compared to both primary Sjögren’s disease patients who did not experience NHL transformation (fold change: 3.72 ± 0.80; *p* = 0.02) and the sicca disease cohort (fold change: 1.81 ± 0.44; *p* = 0.004) (Figure 3C).

Moreover, IL6-R expression directly correlated with miR-155 expression in the salivary glands of patients with primary Sjögren’s disease irrespective of NHL development (R = 0.514; *p* = 0.01) and hypergammaglobulinemia (R = 0.622; *p* = 0.001) (Figure 3A).

## 3. Discussion

This study primarily aimed to identify novel potential biomarkers that could help in stratifying primary Sjögren’s disease patients according to their risk of experiencing lymphoproliferative malignancies.

To date, given its severe impact on their overall survival, the early identification of primary Sjögren’s disease patients at high risk of developing NHL remains the most compelling unmet need for the disease’s management [21,22,23]. Indeed, while numerous prognostic factors—including clinical and laboratory parameters—have been proposed in the context of primary Sjögren’s disease-related lymphoma, only two of them hold the status of primary predictors: persistent salivary gland swelling and mixed cryoglobulinemia, which are biologically linked [8,21,23]. When exploring histological biomarkers, the roles of FS and ectopic lymphoid structures are well established as more relevant histopathological factors in terms of disease severity and risk of lymphoma development [8,21]. In the present study, we found no difference in terms of FS between patients who developed NHL and those who did not. In this scenario, considering the prominent role of B-cells in primary Sjögren’s disease’s pathogenesis, we proposed the upregulation of miR-155, BAFF-R, and IL-6R as potential B-cell activation biomarkers capable of identifying patients who were more likely to experience NHL development in a cohort of primary Sjögren’s disease patients not exhibiting the canonical risk factors. Surely, this could also be of greater potential interest if reproduced in a larger cohort.

Actually, the pathogenic mechanisms fostering the transformation into a lymphoproliferative process remain unclear. Certainly, LSGs are an attractive source of biomarkers for stratification purposes, being the site of the inflammatory process. In this context, the regulatory function of miRNAs may hold a prominent position, considering the potential therapeutic implications. The literature provides a wide range of evidence regarding the role of miRNAs in B-cell function. Dysregulation of miR-155 has been observed in multiple autoimmune diseases [9,10,13], induced by several pathways including the BAFF/BAFF-R signal [15,16,17]. Moreover, increased miR-155 expression has been reported to be associated with more aggressive forms of B-cell lymphomas [14]. Based on this, we studied the roles of miR-155 and the BAFF/BAFF-R axis at the LSG site as a putative tissue biomarker predictor of NHL development in primary Sjögren’s disease.

Notably, the research on miRNAs in primary Sjögren’s disease is relatively new [24,25]. Furthermore, there are several potentially confounding factors to take into account when studying their role in autoimmune diseases. Specifically, regarding primary Sjögren’s disease, several studies have analysed different types of samples, including LSG, peripheral blood mononuclear cells (PBMCs), monocytes, saliva, or exosome from saliva [13]. It is evident that, when analysing a mix of cellular populations, it is mandatory to use caution in the interpretation of miRNAs data, due to possible contamination by microorganism-derived RNA molecules.

In our study, we demonstrated an increased expression of miR-155, BAFF-R, and IL-6R in glandular tissue of primary Sjögren’s disease patients, as assessed by RT-PCR, compared to sicca disease controls. A combined evaluation of miR-155, BAFF-R, and IL-6R expression in glandular tissue has not previously been conducted. Although its expression by individual cytotypes is unknown (immunofluorescence was not performed as confirmation), we can postulate that B-cells might be the primary source of increased expression, as IHC analysis of glandular tissue showed a higher predominance of B lymphocytes and plasma cells in primary Sjögren’s disease patients compared to sicca disease controls, which is strongly consistent with the literature [26,27].

Several studies have focused on the expression of miRNAs in circulating T and B-cells from patients with primary Sjögren’s disease. Specifically, Wang-Renault et al. have recently detected an increased expression level of miR-155 in T lymphocytes of patients with primary Sjögren’s disease compared to healthy controls [28]. Furthermore, a direct correlation between miR-155 expression in PBMCs of primary Sjögren’s disease patients and the Visual Analog Scale (VAS) score for dry eyes has been reported [29]. Investigation of the FoxP3 transcription factor, which is overexpressed in T cells infiltrating salivary glands, has been shown to promote miR-155 expression. Interestingly, an increased level of miR-155 expression has also been revealed in cultured epithelial cells from salivary glands of primary Sjögren’s disease patients compared to controls [26,27]. Shi et al. recently investigated the expression of miR-155 in PBMCs isolated from a cohort of primary Sjögren’s disease patients similar in size to ours, demonstrating a reduced expression of miR-155 in primary Sjögren’s disease patients compared with healthy controls [29]. Conversely, Pauley et al. previously demonstrated an increased expression of miR-155 in PBMCs isolated from primary Sjögren’s disease patients, albeit receiving immunosuppressive therapy [11]. Both studies failed to demonstrate a correlation between miRNA expression and disease characteristics, except the former study, which showed a direct correlation between miR-155 and the VAS score for dry eyes [29]. Recently, Chen et al. reported a wide range of miRNAs being over-expressed in PBMCs of patients with SLE but not changed in those with primary Sjögren’s disease, including miR-148a-3p, miR-152, miR-155, miR-223, miR-224, miR-326, and miR-342 [30]. Therefore, further studies are needed to understand the exact regulatory function of miR-155.

In this context, our study demonstrates that minor salivary gland tissue of primary Sjögren’s disease patients is enriched with miR-155 compared to sicca disease, and its expression positively correlates with the degree of inflammatory infiltration in terms of FS as well as hypergammaglobulinemia, suggesting a possible prominent role in B-cell activation in primary Sjögren’s disease.

Nevertheless, the unique feature of primary Sjögren’s disease is its association with B-cell proliferation [31,32]. The available evidence explaining the mechanisms by which polyclonal B-cells with autoimmune behaviour acquire monoclonal characteristics is still limited. The inflammatory and lymphoproliferative process involving mucosal associate lymphoid tissue sites in other diseases, such as gastric infection by Helicobacter pylori [33,34], provides an important model for studying the characteristics of B-cell proliferation in primary Sjögren’s disease. Analysing the primary Sjögren’s disease cohort, patients who developed NHL showed higher miR-155 expression at the LSG level than primary Sjögren’s disease patients who did not experience NHL transformation. MiR-155 is known to play a key role in the pathogenesis of B-cell lymphomas [5,14] and Rai et al. demonstrated that miR-155 expression in peripheral blood-derived B-cells of DLBCL patients segregates with certain molecular subtypes of the disease [35,36].

Furthermore, proteins related to cell proliferation (cyclin B1, cyclin D1, CDK4) were found to be inhibited in DLBCL due to downregulation of miR-155, which in turn led to an increased expression of pro-apoptotic proteins Bax/Bcl2 and caspase 3 [37]. Although further validation in larger cohorts is needed, these findings place miR-155 in a central position linking its role in B-cell biology to B-cell lymphoproliferation. Based on the aforementioned findings, miR-155 may serve as a promising biomarker characterizing primary Sjögren’s disease patients at the highest risk of malignant transformation as early as the diagnosis stage, also arising as a potential therapeutic target.

Given the prominent role of B-cells in primary Sjögren’s disease, the role of BAFF, which promotes B-cell activation and belongs to the TNF family [38], is consequently remarkable. Indeed, BAFF plays an important role in the pathogenesis of the disease, as it is associated with the breakdown of tolerance and the production of autoantibodies [38,39,40]. Furthermore, BAFF is responsible for the abnormal distribution of different subsets of PB B-cells in primary Sjögren’s disease [41]. BAFF is the only known cytokine capable of activating the BAFF receptor, which is expressed by circulating B- and T cells [42]. Mouse models have shown a crucial role for BAFF in B-cell survival, as BAFF or BAFF-R gene knockout mice exhibit reduced numbers of peripheral B-cells [43]. In this scenario, miRNAs and BAFF/BAFF-R system analysis could play a decisive role, alongside previously detected epidemiological, clinical, laboratory, and histological-adverse predictive markers [8,44]. Moreover, Kapsogeorgou et al. recently described the low expression of miR200b-5p in the minor salivary glands of primary Sjögren’s disease patients as a plausible pathogenetic mechanism-related factor impacting the development of primary Sjögren’s disease-associated NHL [45].

Thus, considering the promising role of BAFF in the pathogenesis of the disease, our study evaluated the expression of its receptor (BAFF-R) by RT-PCR at the LSG site. Of note, BAFF-R was observed to be significantly higher in the LSGs of patients who experienced lymphomagenesis compared to those of primary Sjögren’s disease patients who did not develop NHL, and directly correlated with in situ miR-155 expression. These findings are consistent with previous studies that demonstrated the presence of BAFF in LSG epithelial cells from primary Sjögren’s disease patients [46,47]. Furthermore, the BAFF/BAFF-R system has been shown to potentially be associated with focal lymphocytic infiltration in the LSGs of primary Sjögren’s disease patients, serving as a mechanism for the possible establishment of GC-like ectopic structures and the progression of the disease in some patients [39]. However, this is the first study linking BAFF-R upregulation with malignant transformation in primary Sjögren’s disease. Indeed, the BAFF-R serves as one of the main pro-survival receptors in B-cells, and its aberrant expression has been reported in NHL patients by many studies focusing on the distribution of BAFF-R among distinct B-cell subsets and their neoplastic counterparts [48,49].

Thus, the upregulation of BAFF-R in the cohort of primary Sjögren’s disease patients who experienced neoplastic evolution and its direct correlation with miR-155 levels indicate the BAFF/BAFF-R axis as one of the potential key signalling pathways in miR-155-mediated B-cell proliferation in primary Sjögren’s disease.

Although data on IL-6R inhibition in primary Sjögren’s disease are not encouraging [18], the IL-6 axis can be crucial in the pathobiology of the disease by acting as a B-cell growth and differentiation factor, promoting T helper 17 (Th17) expansion, and exerting polyclonal activation of B-cells, which in turn are capable of producing IL-6 [18,19,20]. Moreover, the LSGs of patients suffering from primary Sjögren’s disease have been shown to contain IL-6, which has also been found to be overexpressed in the serum, saliva, and tear fluid of primary Sjögren’s disease patients [50,51,52].

To the best of our knowledge, there are no published studies assessing the expression of IL-6R in salivary glands of patients with primary Sjögren’s disease.

Interestingly, our study proved that minor salivary gland tissue of primary Sjögren’s disease patients is enriched with IL-6R compared to sicca disease controls, and its expression positively correlates with the expression of miR-155 as well as with hypergammaglobulinemia, confirming its role in B-cell polyclonal activation.

Furthermore, IL-6R salivary gland tissue expression was upregulated in patients who developed NHL during follow-up compared to those who did not experience lymphomagenesis. It is known that the IL-6 signalling complex is involved in haematological malignancies, markedly in B-cell-derived and plasma cell tumours [53,54,55,56].

Thus, it is conceivable that it, along with the miR-155 and BAFF/BAFF-R axis, may have a critical function in the B-cell aberrant activation leading to malignant transformation.

Although this is the first study evaluating miR155 combined with IL6/BAFF-R expression in salivary glands as B-cell activation biomarkers related to lymphomagenesis in pSS, its retrospective design and the limited sample size of patients experiencing NHL evolution are limitations that should be considered. Therefore, further investigation using a larger cohort is needed to confirm the prognostic power of our findings.

In conclusion, epigenetic modulation through miR-155 induction may play a crucial role in the aberrant activation of B-cells in primary Sjögren’s disease, promoting an increasing risk of NHL development throughout the disease’s course and arising as a putative novel biomarker of patients’ stratification.

## 4. Materials and Methods

### 4.1. Patient Selection

Twenty-four patients with primary Sjögren’s disease, diagnosed according to current classification criteria [57,58,59] and with available paraffin-embedded (FFPE) tissue blocks from LSG biopsies performed at diagnosis, were included in this single-centre study at the Fondazione Policlinico Universitario A. Gemelli IRCCS in Rome.

Among the included dataset, 5 FFPE tissue blocks belonged to primary Sjögren’s disease patients who developed B-cell NHL during the 14-year follow-up from 2009 to 2023, while 19 FFPE tissue blocks belonged to patients with primary Sjögren’s disease who had not developed NHL during a similar follow-up period; they were included using a random matching approach [60] for gender and age at diagnosis. A comparison group of 20 FFPE tissue blocks from patients with sicca syndrome, who were not diagnosed with primary Sjögren’s disease after their LSG biopsies, was included.

### 4.2. Minor Salivary Glands Biopsy

Each enrolled patient underwent an LSG biopsy under local anaesthesia, following the technique described by Daniels [61]. All LSG samples were fixed in 10% neutral buffered formalin, embedded in paraffin, and routinely stained with haematoxylin and eosin (H&E). The samples were examined using a light microscope (Leica DM2000) by two independent evaluators who were unaware of the clinical and immunological characteristics of the patients. Focus score (FS) evaluation was performed to assess minor salivary gland inflammation according to Greenspan and Daniels’ method [62].

### 4.3. Clinical and Immunological Assessment of Patient Cohorts

Upon study entry, demographic, clinical, and immunological parameters were recorded for each patient. In particular, laboratory tests included detection of serum cryoglobulins, serum levels of complement factors (C3 and C4), immunoglobulin levels, erythrocyte sedimentation rate (ESR), and C-reactive protein (CRP). All patients were tested for peripheral blood (PB) IgA and IgM rheumatoid factor (RF) isotypes (Orgentec Diagnostika, Dundee, UK), antinuclear antibodies (ANA), and anti-Ro (SSA) and anti-La (SSB) antibodies. The reference cut-off levels were 20 U/mL for IgA and IgM-RF and 10 AU/mL for anti-Ro(SSA) and La(SSB) antibodies.

After diagnosis, each primary Sjögren’s disease patient was treated according to the current recommendations for primary Sjögren’s disease management [63,64,65], and followed-up with every 6 months in an outpatient clinical setting. Particularly, at each study visit, a comprehensive clinical examination was performed for primary Sjögren’s disease patients to assess whether the disease was limited to glandular involvement or whether extra-glandular manifestations (EGMs) were present [3,4].

Disease activity was assessed according to the EULAR Sjögren’s Disease Activity Index (ESSDAI) as follows: ESSDAI < 5: low activity; 5 ≤ ESSDAI ≤ 13: moderate activity; and ESSDAI ≥ 14: high activity [66,67].

Follow-up of primary Sjögren’s disease patients lasted until the development of NHL, or until the last available follow-up visit for primary Sjögren’s disease patients who did not develop NHL (at least 10 years follow-up).

The study protocol was approved by the local Ethics Committee of the Catholic University of the Sacred Heart (approval no. 6334/15). All subjects signed the provided informed consent forms.

### 4.4. Immunohistochemistry of CD20 and CD138 in Labial Salivary Gland Tissue

LSG samples obtained from primary Sjögren’s disease and sicca disease patients were tested through immunohistochemistry (IHC) for B-cell and plasma cell detection, using CD20 mouse antihuman monoclonal antibody (clone L26) and CD138 mouse antihuman monoclonal antibody (clone MI15) (all from Leica Biosystem, Newcastle, UK), respectively, by immunostainer BOND MAX III (Leica), following the published protocol [68].

Slides were evaluated by an experienced pathologist using a light microscope (Leica DM 2000), and all samples were assessed using a numerical score based on the number of CD20 and CD138 positive cells (two different fields in each section), with a score of 0 indicating no positive cells, 1 indicating < 10% positive cells, 2 indicating 10–50% positive cells, and 3 indicating > 50% positive cells [69,70].

### 4.5. miR-155, BAFF-R, and IL-6R Expression in Labial Salivary Gland Biopsy by qPCR

Total RNA was isolated from the LSGs of 24 primary Sjögren’s disease patients and 20 sicca disease controls using the miRneasy kit (Qiagen, Hilden, Germany). The miScript Reverse Transcription kit (Qiagen, Hilden, Germany) was used for cDNA preparation, and the miScript PreAMP PCR kit (Qiagen, Hilden, Germany) was used for cDNA pre-amplification following the manufacturer’s instructions. The expression of miR-155 and U1 (housekeeping control) were evaluated using the SYBR Green method using human miR-155 (MS00003605)- and U1 (MS00013986)-specific primers (Qiagen, Hilden, Germany). The miScript miRNA PCR array enables SYBR Green-based real-time PCR analysis using the Biorad iQ5 real-time PCR system as follows: 95 °C for 15 min; 40 cycles of 94 °C for 15 s; 55 °C for 30 s; and 70 °C for 30 s. The relative expression was calculated using the ΔΔCt method (relative gene expression = 2(ΔCttest − ΔCtcontrol)) and is presented as fold increase relative to control.

The iScriptTM cDNA Synthesis Kit (Bio Rad Laboratories, Hercules, CA, USA) was used for cDNA preparation. Expression of BAFF-R, IL-6R, and GAPDH (housekeeping control) were assessed by the SYBR Green PCR Kit (Bio Rad Laboratories, Hercules, CA, USA) following the manufacturer’s protocol. For the semi-quantitative determination of the expression of human BAFF-R, IL-6R, and control GAPDH in LSG tissue, ΔCt values were generated after subtraction from the gene of interest with the Ct value of control (GAPDH).

### 4.6. Statistical Analysis

Statistical analysis was performed using SPSS V. 20.0 (SPSS. Chicago, IL, USA) and Prism software (GraphPad-9, San Diego, CA, USA). Categorical and quantitative variables were described as frequencies, percentage, and mean ± standard error of the mean (SEM). Data on demographic, immunological, and clinical features were compared between patients by non-parametric one-way ANOVA, the Mann–Whitney U test, or the χ2 test, as appropriate. Spearman’s rank correlation test was used for correlation in all analyses. A receiver operating characteristics (ROC) curve analysis of miR-155 fold change was performed to obtain relevant thresholds, allowing the prediction of disease status as well as neoplastic evolution at the baseline. The optimal cut-off point was determined to yield the maximum corresponding sensitivity and specificity. For all the analyses, a *p* < 0.05 was considered statistically significant and all tests were two-tailed, unless otherwise indicated.

## Figures and Tables

**Figure 1 ijms-25-03259-f001:**
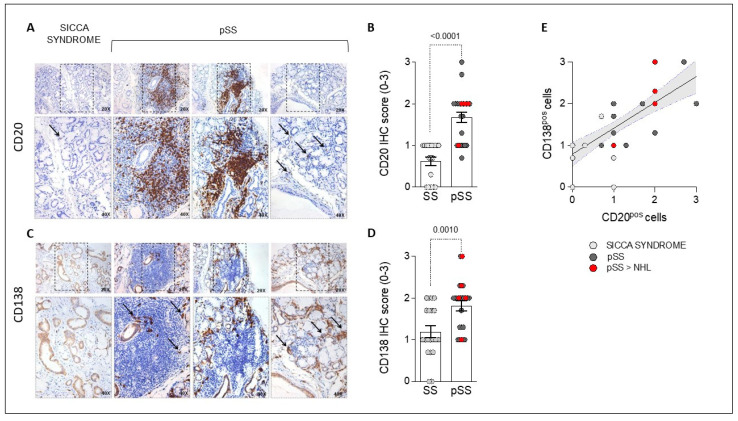
(**A**–**E**). IHC staining for CD20 and CD138 in LSG samples from sicca syndrome and primary Sjögren’s disease (pSS) patients. Example photos of CD20 (DAB) (**A**) and CD138 (DAB) (**C**) staining of LSG biopsies from patients with sicca syndrome or pSS (magnification x20 and magnification x40 in the corresponding inset). Black arrows indicate CD20pos cells and CD138pos cells in the corresponding inset. LSG IHC score for CD20pos cells (**B**) and CD138pos cells (**D**) in sicca syndrome and pSS patients. Correlation between LSG IHC score for CD138pos cells and CD20pos cells in the enrolled cohorts, Spearman rank correlation test. (**E**) IHC, immunohistochemistry; LSG, labial salivary glands; pSS, primary Sjögren’s disease; DAB, 3,3-diaminobenzidine; NHL, non-Hodgkin’s lymphoma.

**Figure 2 ijms-25-03259-f002:**
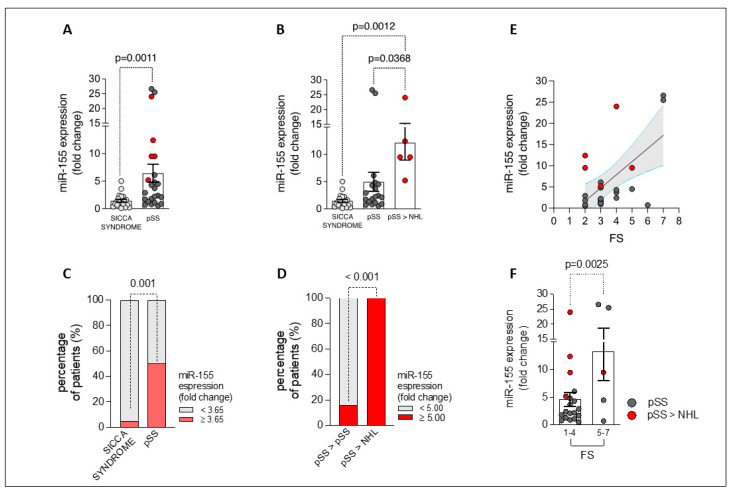
(**A**–**E**) miR-155 expression in the LSG samples from sicca syndrome and primary Sjögren’s disease (pSS) patients. LSG miR-155 expression (fold change) in sicca syndrome and pSS patients (**A**) stratified based on NHL development (**B**). Percentage of patients with sicca syndrome and pSS having a < or ≥3.65- (**C**) and < or ≥5.00-fold (**D**) increase of miR-155 expression based on NHL development. Correlation between LSG miR-155 expression (fold change) and FS in pSS cohort, Spearman rank correlation test (**E**). LSG miR-155 expression (fold change) in pSS patients divided by the degree of FS (**F**). LSG, labial salivary glands; SS, sicca syndrome; pSS, primary Sjögren’s disease; NHL, non-Hodgkin’s lymphoma; miR-155, microRNA 155; FS, lymphocytic focus score (foci/4 mm^2^).

**Figure 3 ijms-25-03259-f003:**
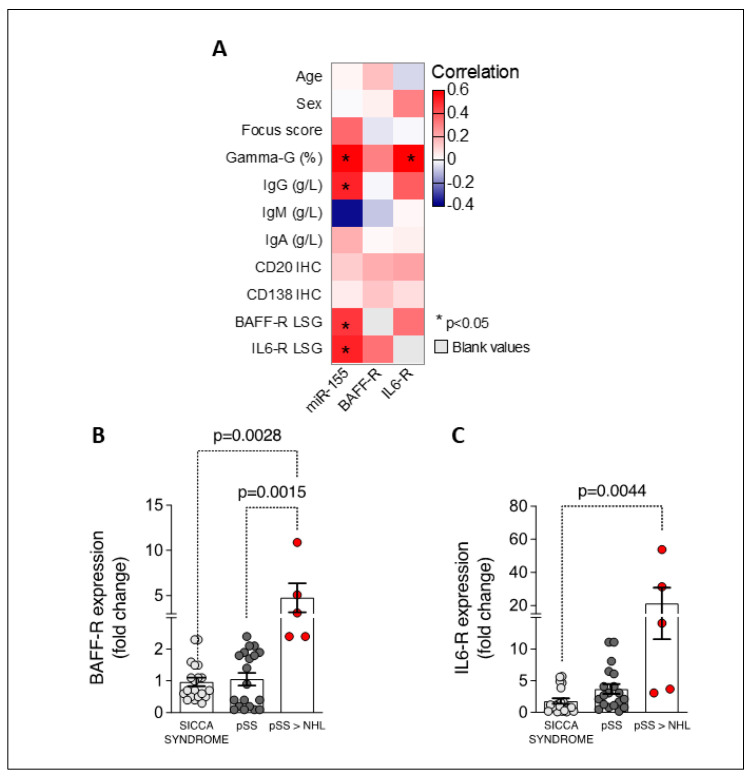
(**A**–**C**). BAFF-R and IL-6R expression in the LSGs of sicca syndrome and primary Sjögren’s disease (pSS) patients. Heatmap showing correlation between LSG miR-155, BAFF-R, and IL-6R expression (fold change) and demographic, histologic, and laboratory parameters in pSS cohort (**A**). LSG BAFF-R expression (fold change) in sicca syndrome and pSS patients stratified based on NHL development (**B**). LSG IL-6R expression (fold change) in sicca syndrome and pSS patients stratified based on NHL development (**C**). LSG, labial salivary glands; BAFF-R, BAFF receptor; IL-6R, IL-6 receptor; pSS, primary Sjögren’s disease; NHL, non-Hodgkin’s lymphoma.

**Table 1 ijms-25-03259-t001:** Demographic, clinical, immunological, and histopathological characteristics of the enrolled cohort.

Variable	Sicca Disease (n = 20)	pSS Cohort (n = 24)	*p*	pSS without NHL Evolution (n = 19)	pSS with NHL Evolution (n = 5)	*p* *	*p* **	*p* ***
Gender, female (%)	19 (95.0)	21 (87.5)	0.39	17 (89.4)	5 (100)	0.45	0.52	0.62
Age (years)	52.5 (40.3–57.3)	54.5 (47.3–64.8)	0.10	53.0 (46.0–61.0)	67.0 (54.5–77.0)	**0.04**	0.33	**0.02**
Disease duration (years)	-	10.5 ± 3.4	-	12.3 ± 1.4	7.3 ± 3.1	0.13	-	-
Lymphocytic focus score (foci/4 mm^2^)	0	3.5 ± 0.3	**<0.0001**	3.6 ± 0.4	3.2 ± 0.6	0.64	**<0.0001**	**<0.0001**
ANA	4 (20)	24 (100)	**<0.0001**	19 (100)	5 (100)	1	**<0.0001**	**<0.0001**
Anti-Ro/SSA, n (%)	0 (0)	22 (92)	**<0.0001**	18 (95)	4 (75)	0.18	**<0.0001**	**<0.0001**
Anti-La/SSB, n (%)	0 (0)	20 (83)	**<0.0001**	17 (89)	3 (60)	0.13	**<0.0001**	**0.0003**
RF-IgA^pos^, n (%)	0 (0)	1 (0.04)	0.93	1 (0.05)	0 (0)	0.96	0.92	1
RF-IgM^pos^, n (%)	0 (0)	3 (0.13)	0.87	3 (0.16)	0 (0)	0.93	0.86	1
IgG (mg/dL)	1147.7 ± 59.8	1247.7 ± 79.6	0.34	1184.6 ± 60.7	1468.8 ± 291.2	0.14	0.67	0.09
IgM (mg/dL)	138.2 ± 12.6	119.5 ± 11.9	0.29	132.6 ± 12.8	69.4 ± 16.7	**0.03**	0.76	**0.02**
IgA (mg/dL)	193.9 ± 20.4	228.8 ± 26.7	0.32	188.9 ± 20.1	380.6 ± 73.9	**0.002**	0.86	**0.002**
Gamma-globulin (%)	14.3 ± 0.6	17.1 ± 0.9	**0.02**	15.9 ± 0.8	21.4 ± 1.8	**0.006**	0.12	**0.0001**
ALC (×10^9^/L)	2319.9 ± 304.3	1928.1 ± 77.4	0.18	1685.6 ± 90.8	1889.6 ± 126.7	0.29	0.06	0.49
C3 (mg/dL)	109.3 ± 4.3	111.9 ± 3.8	0.65	112.4 ± 4.7	109.5 ± 4.1	0.76	0.63	0.98
C4 (mg/dL)	21.7 ± 1.6	25.0 ± 2.2	0.25	22.6 ± 1.8	24.5 ± 6.7	0.70	0.71	0.54
CRP (mg/L)	2.0 ± 0.6	1.9 ± 0.4	0.89	1.7 ± 0.3	3.1 ± 2.7	0.34	0.66	0.54
ESR (mm/1st h)	14.1 ± 3.1	21.0 ± 2.8	0.11	19.3 ± 3.1	26.5 ± 6.8	0.31	0.24	0.09
ESSDAI	-	2.8 ± 0.3	-	6.5 ± 2.7	8.7 ± 4.2	0.70	-	-

*p* *: pSS without NHL evolution vs. pSS with NHL evolution; *p* **: sicca disease vs. pSS without NHL evolution; *p* ***: sicca disease vs. pSS with NHL evolution. Age is shown as median (25th percentile–75th percentile). Other variables are shown as mean ± SEM or n (%), as appropriate. pSS, primary Sjogren disease; NHL, non-Hodgkin’s lymphoma; ANA, antinuclear antibody; anti-Ro/SSA, anti-Ro/Sjögren’s disease-related antigen A antibody; anti-La/SSB, anti-La/Sjögren’s disease-related antigen B antibody; RF, Rheumatoid Factor; IgA, immunoglobulin A; IgM, immunoglobulin M; IgG, Immunoglobulin G serum levels; IgM, Immunoglobulin M serum levels; IgA, Immunoglobulin A serum levels; ALC, Absolute Lymphocyte Count; CRP, C-reactive protein; ESR, erythrocyte sedimentation rate; ESSDAI, EULAR Sjögren’s Disease Activity Index. Bold values indicate *p* < 0.05.

## Data Availability

Data is contained within the article and Appendix A.

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
