# Peer review of "B-Cell Activation Biomarkers in Salivary Glands Are Related to Lymphomagenesis in Primary Sjögren’s Disease: A Pilot Monocentric Exploratory Study"

_ijms, 2024, doi:10.3390/ijms25063259_

Round 1
Reviewer 1 Report (New Reviewer)
Comments and Suggestions for Authors
Bruno et al studied B-Cell Activation Biomarkers in Salivary Glands in patients with primary Sjögren's Disease. Performed expression analysis of miR-155/BAFF-R and IL-6R in pSS with and without NHL.
1. Major concern is very limited sample size in NHL. Total number of patients in NHL is only 5. This is not enough for any statistical analysis including ROC.Is authors can be do power analysis for this?
2. Authors need more numbers of samples to validate the results.
3. Another important flaw is negative association of focus score and miR-155 i.e most of NHL patients less focus score and more miRNA (figure 2F). On other authors showing positive correlation between this (figure 2E). This is controversial to above statement.
4. Focus score is not significant for expression of BAFF and IL-6R. Literature reported strong correlation between these cytokines.
Minor comments
1. pSS>pSS; pSS>NHL in table need to be modified for better clarity
2. Lot of demographic details is missing in table-1. Only laboratory parameters is mentioned and lack of clinical information.
Author Response
Please see the attachment.

Reviewer 2 Report (New Reviewer)
Comments and Suggestions for Authors
Thank you for an excellent piece of important work in pSS
However, the relationship of complement C3 and C4 was not completely explored given that low C4 has been consistently shown as one of the important predictors in lymphoma development; for example https://www.ncbi.nlm.nih.gov/pmc/articles/PMC8754462/
Can the authors explain their findings where C4 does not seem to have any effect?
Also, what does pSS>pSS mean in Table 1?
Was absolute lymphocyte counts considered in the study?
Author Response
Please see the attachment.

Reviewer 3 Report (Previous Reviewer 3)
Comments and Suggestions for Authors
The authors responded to the previous concerns, suggestions, and questions raised by reviewers. The revised version of the manuscript is improved and contains relevant information
Round 2
Reviewer 1 Report (New Reviewer)
Comments and Suggestions for Authors
The authors have satisfactorily clarified the questions raised. I have suggested some points to improve the manuscript
1. Consider to include the limitations of the study in the discussion
2. In Figure 3a. IL-6R and BAFF-R do not correlate with each other. Need some clarification.
3. Mention the title as “pilot study” because the samples in the NHL group are limited
Author Response
Please see the attachment.

This manuscript is a resubmission of an earlier submission. The following is a list of the peer review reports and author responses from that submission.
Round 1
Reviewer 1 Report
Comments and Suggestions for Authors
The study is well researched and presented concisely: the expression of MiR-155 and BAFF-R in Primary Sjögren's syndrome. Specifically the over-expression of MiR-155 in lymphoma and pSS. The topic is relevant to the field but would not be considered original as the theory has been proven in other similar diseases. This would be the first known study that proves this theory in biopsy samples taken from pSS patients. The conclusions consistent with the evidence and arguments are presented. The references are appropriate.
I have only one comment for the authors.
The study included 20 samples from each of the study category. In their opinion is this a significant enough for the conclusions presented? The introduction doesn't indicate the prevalence rate of the disease.
Specific improvements regarding the methodology.
Micro Dissecting specific cells from the biopsy samples to perform the qPCR would beneficial as this would decrease any signal coming from other surrounding normal cells.
Author Response
The study is well researched and presented concisely: the expression of MiR-155 and BAFF-R in Primary Sjögren's syndrome. Specifically the over-expression of MiR-155 in lymphoma and pSS. The topic is relevant to the field but would not be considered original as the theory has been proven in other similar diseases. This would be the first known study that proves this theory in biopsy samples taken from pSS patients. The conclusions consistent with the evidence and arguments are presented. The references are appropriate.
I have only one comment for the authors.
The study included 20 samples from each of the study category. In their opinion is this a significant enough for the conclusions presented? The introduction doesn't indicate the prevalence rate of the disease.
Specific improvements regarding the methodology.
Micro Dissecting specific cells from the biopsy samples to perform the qPCR would beneficial as this would decrease any signal coming from other surrounding normal cells.
We thank the Reviewer for the kind comment and propose to add in the Introduction section a sentence with regard to the prevalence of Primary Sjögren's syndrome (lines 36-37).
We know that Primary Sjögren's syndrome is far from a rare disorder, affecting 0.5% to 1.0% of the population, as well as we are aware that the number of patients enrolled represents a limitation of the study. However, due to the retrospective nature of the study, we only wanted to enrol patients whose clinical history and tissue quality we could be certain of. To conclude, we believe that the number of patients can be sufficient.
About the additional comment on the qPCR methodology, we absolutely agree with the suggestion. In fact, in the Discussion section, we stated that the miR-155 and BAFF-R expression in glandular tissue by individual cytotypes is unknown, given that immunofluorescence was not performed as confirmation. (lines 330-336).
This is definitely a caveat of the study. However, we can postulate that B-cells might be the primary source of increased expression as IHC analysis of glandular tissue showed a higher predominance of B lymphocytes and plasmacells in pSS patients compared to SS, which is strongly consistent with the literature. (lines 330-336).
Reviewer 2 Report
Comments and Suggestions for Authors
In this study, the authors aim to explore the miR-155 and BAFF-R expression in the labial salivary glands 18 (LSG) from patients with pSS and identify novel biomarkers of NHL development. Using RT-qPCR and IHC, they detected local inflammation and miR-155/BAFFR expression in labial salivary gland tissue. Although the findings are interesting, more in-depth analysis is needed.
11. I noticed that among the pSS patients, 21 were females (87.5%). Is Primary Sjögren's Syndrome (pSS) more common in women?
22. Related to Figure 1C and 1D, the horizontal axis icons SS and pSS should be separated. If not, it will confuse the readers. Also, the quality of the figures should be improved.
33. Related to Figure 3, the results showed that BAFFR mRNA expression was higher in pSS patients than in SS patients. Did the authors test BAFFR protein expression in the LSG from SS and pSS patients using Western blot? Since BAFF is the only known cytokine capable of activating the BAFFR, did the authors detect BAFF expression?
44. At line 263 and 264, the authors concluded that BAFFR expression positively correlated with miR-155 expression in LSG regardless of NHL development. Can the authors explain the link between miR-155 and BAFFR? Does miR-155 affect the expression of BAFFR directly or indirectly? What is the target of miR-155?
55. The authors only tested miR-155 expression in LSG. What is the expression of miR-155 in PBMC or CD14 monocytes of SS and pSS patients?
Author Response
In this study, the authors aim to explore the miR-155 and BAFF-R expression in the labial salivary glands 18 (LSG) from patients with pSS and identify novel biomarkers of NHL development. Using RT-qPCR and IHC, they detected local inflammation and miR-155/BAFF-R expression in labial salivary gland tissue. Although the findings are interesting, more in-depth analysis is needed.
- I noticed that among the pSS patients, 21 were females (87.5%). Is Primary Sjögren's Syndrome (pSS) more common in women?
We thank the Reviewer for the kind comment. The enrolled cohort is in line with the literature, as the Primary Sjögren's Syndrome (pSS) is an autoimmune disease that occurs predominately in women over men, with an estimated female to male ratio of 9–14 to 1. (Brandt JE, Priori R, Valesini G, Fairweather D. Sex differences in Sjögren's syndrome: a comprehensive review of immune mechanisms. Biol Sex Differ. 2015 Nov 3;6:19. doi: 10.1186/s13293-015-0037-7. PMID: 26535108; PMCID: PMC4630965; Ramírez Sepúlveda JI, Kvarnström M, Brauner S, Baldini C, Wahren-Herlenius M. Difference in clinical presentation between women and men in incident primary Sjögren's syndrome. Biol Sex Differ. 2017 May 12;8:16. doi: 10.1186/s13293-017-0137-7. PMID: 28507729; PMCID: PMC5427625)
- Related to Figure 1C and 1D, the horizontal axis icons SS and pSS should be separated. If not, it will confuse the readers. Also, the quality of the figures should be improved.
We thank the Reviewer for the kind comment. As suggested, we improved the quality of Figure 1.
- Related to Figure 3, the results showed that BAFF-R mRNA expression was higher in pSS patients than in SS patients. Did the authors test BAFF-R protein expression in the LSG from SS and pSS patients using Western blot? Since BAFF is the only known cytokine capable of activating the BAFF-R, did the authors detect BAFF expression?
We thank the Reviewer for the appropriate comment regarding the methodology. Indeed, the results showed that minor salivary glands of patients with Primary Sjögren's Syndrome was enriched of BAFF-R (fold change: 2.55 ± 0.68) compared to patients suffering from sicca syndrome. As reported in the Methods section, we did not use Western blot to prove this on the protein level. We isolated total RNA from labial salivary glands of enrolled patients and do qPCR following the specific protocol to detect miRNA expression (lines 141-158). Thus, we have come to an indirect conclusion about protein abundance. BAFF expression evaluation was not performed.
- At line 263 and 264, the authors concluded that BAFF-R expression positively correlated with miR-155 expression in LSG regardless of NHL development. Can the authors explain the link between miR-155 and BAFF-R? Does miR-155 affect the expression of BAFF-R directly or indirectly? What is the target of miR-155?
We thank the Reviewer for the kind comment regarding the link between miR-155 and BAFF. We observed a direct correlation between their expression in labial salivary glands of patients suffering from Primary Sjögren's Syndrome (pSS). We think It is reasonable to assume that there is a strict biological link between the miR-155 and BAFF/BAFF-R axis expression in the B-cell activation in pSS, although the are no data published yet. Indeed, Alivernini et al. found that BAFF, which is up-regulated in synovial fluid compared to peripheral blood of patients wih Rheumatoid Arthritis, is able to significantly increase miR-155 expression in CD19+ cells after stimulation. Similarly, Cui et al. reported that BAFF significantly upregulated the expression of miR-155 in patients with chronic lymphocytic leukemia, which in turn is generally associated with a more aggressive disease. Thus, since BAFF-R binds exclusively to BAFF, and considering that there are no studies addressing its relation to miR-155 in autoimmune diseases, we believe that this could be an interesting novelty proposed by the study, assuming that miR-155 and BAFF/BAFF-R axis could enhance each other in the aberrant B-cell activation in pSS.
- The authors only tested miR-155 expression in LSG. What is the expression of miR-155 in PBMC or CD14 monocytes of SS and pSS patients?
We thank the Reviewer for the comment. In the present study, we demonstrated an increased expression of miR-155, as well as of BAFF-R, in labial salivary glands of pSS patients compared to SS controls. The novelty of the study is represented by the fact that a combined evaluation of miR-155 and BAFF-R expression in glandular tissue has not previously been conducted. As mentioned in the Discussion section, several studies focused on the expression of miRNAs in circulating T and B cells from patients with pSS. Specifically, Wang-Renault et al. have detected an increased expression levels of miR-155 in T lymphocytes of patients with pSS compared to healthy controls. Interestingly, Shi et al. recently investigated the expression of miR-155 in PBMCs isolated from a group of pSS patients similar in size to our cohort, demonstrating a reduced expression of miR-155 in pSS compared to healthy controls. Conversely, Pauley et al. previously have demonstrated an increased expression of miR-155 in PBMCs isolated from pSS patients, albeit receiving immunosuppressive therapy.
About CD14+ monocytes, it is well known its pro-inflammatory phenotype in pSS, being proposed to contribute to salivary gland inflammation. Specifically, they secrete increased levels of pro-inflammatory cytokines such as interleukin (IL)-6 and B cell-activating factor (BAFF) upon stimulation, express type I interferon (IFN)-regulated genes, reduced NFκB inhibitor and show decreased function in phagocytosis of apoptotic cells.
Williams et al published the most important study on peripheral blood CD14+ monocytes and miRNAs in pSS, reporting an up-regulation of miR-300, miR-609, miR-3162-3p, and miR-4701-5p, in pSS patients compared to healthy controls. We did not include this study in the Discussion section since the authors did not investigate the miR-155 expression.
However, we implemented the Discussion section to shed light on the studies addressing the role of miRNAs in pSS which have been conducted so far.
Reviewer 3 Report
Comments and Suggestions for Authors
The manuscript describes the roles of miRNA-155 and BAFFR in Sjøgren's syndrome-associated B cell lymphoma. There are several points that the authors are encouraged to go through during the revision.
1. pSS is introduced as primary Sjogren's syndrome. It would be beneficial to elaborate on what other options than "primary" exist (Secondary?) and what is the difference.
2. SS is introduced as sicca syndrome. It is written marginally confusing SS vs pSS, which has very different meanings here in the manuscript. One option is to avoid abbreviations for pSS and SS, or at least for SS. It is even more confusing because "SS" in the literature sometimes means "Sjogren's syndrome" as well.
3. There is a gradual shift from the "Sjogren's syndrome" to "Sjøgren's disease" definition in some countries. If authors prefer to use the classical word "syndrome" it is understandable. It might be useful though to mention at least once that the name "Sjogren's disease" exists and means the same.
4. Materials and Methods. Kindly provide detailed information for reagents used (catalog number, country, city, producer).
5. Figure 1A, B. Is it possible to include a scale bar for microscopy-based images?
6. Figure 3. Kindly check the shifted lines. Right top, "SS, pSS>pSS, pSS>NHL". Moreover, kindly check if "pSS>pSS" is correct and whether it should be "SS>pSS"
7. Several mandatory fields are not filled in. It should be fixed during the revision.
Author Contributions:
Funding:
Institutional Review Board Statement:
Informed Consent Statement:
Conflicts of Interest:
Author Response
The manuscript describes the roles of miRNA-155 and BAFF-R in Sjøgren's syndrome-associated B cell lymphoma. There are several points that the authors are encouraged to go through during the revision.
- pSS is introduced as primary Sjogren's syndrome. It would be beneficial to elaborate on what other options than "primary" exist (Secondary?) and what is the difference.
We thank the Reviewer for this comment that improve the quality of the manuscript. Indeed, we modified the text as suggested, explaining in the Introduction section what the secondary Sjogren syndrome is as well as their relative differences. (lines 34-36).
- SS is introduced as sicca syndrome. It is written marginally confusing SS vs pSS, which has very different meanings here in the manuscript. One option is to avoid abbreviations for pSS and SS, or at least for SS. It is even more confusing because "SS" in the literature sometimes means "Sjogren's syndrome" as well.
We thank the Reviewer for this comment that ameliorates the style of the manuscript. Indeed, we modified the text as suggested, avoiding the abbreviations for sicca syndrome.
- There is a gradual shift from the "Sjogren's syndrome" to "Sjøgren's disease" definition in some countries. If authors prefer to use the classical word "syndrome" it is understandable. It might be useful though to mention at least once that the name "Sjogren's disease" exists and means the same.
We thank the Reviewer for this comment that improve the quality of the manuscript. Indeed, in the Introduction section we highlighted the systemic nature of the condition, with almost each organ that can be potentially affected, thus explaining the gradual shift to Sjogren's disease definition. (lines 39-41).
- Materials and Methods. Kindly provide detailed information for reagents used (catalog number, country, city, producer).
We thank the Reviewer for the comment. We provided all reagents informations.
- Figure 1A, B. Is it possible to include a scale bar for microscopy-based images?
We thank the Reviewer for the comment. A scale bar within the microscopy-based image can not be included due to the setting of the instruments used for photo capturing.The detailed magnification scale was included in each figure legend.
- Figure 3. Kindly check the shifted lines. Right top, "SS, pSS>pSS, pSS>NHL". Moreover, kindly check if "pSS>pSS" is correct and whether it should be "SS>pSS"
We thank the Reviewer for the kind comment. As suggested, we improved the quality of Figure 3.
- Several mandatory fields are not filled in. It should be fixed during the revision.
We thank the Reviewer for this kind comment. We filled in all the mandatory fields.
Reviewer 4 Report
Comments and Suggestions for Authors
In this study, the authors aimed to explore the expression of BAFF receptor (BAFF-R) and miR-155 in labial salivary gland tissues of patients with Sjögren`s syndrome (pSS), and the correlation with disease outcome. The strength of the study is that the authors were able to follow up their patients for up to 10 years, enabling them to identify patients who developed lymphoma. With trying to identify biomarkers for lymphomagenesis in patients with pSS, the authors address here an important question of high relevance for patients and clinicians. The novel aspect of the study is the higher expression of miR-155 in tissues of patients with pSS compared to control patients. However, the weakness of this study is its sole descriptive characteristic, the lack of any experimental validation for the role of miR-155 in lymphomagenesis in Sjögren`s Syndrome, and the low number (n=4) of patients who developed lymphoma.
Please find below the specific comments:
Major comments:
In the first sentence of the discussion section the authors mention that “…this study primarily aimed to identify novel potential biomarkers that could help stratifying pSS patients according to their risk of experiencing lymphoproliferative malignancies”. To identify novel biomarkers, one would expect to choose a broader experimental approach rather than pre-selecting two factors (miR-155, BAFF-R).
Among the 24 patients with pSS, only 4 patients developed lymphoma. Among these 4 patients, 2 patients exhibit similar fold changes in expression levels for BAFF-R and miR-155 overlap with pSS patients without lymphoma. Therefore, the differences in data (Figs. 2 and 3) are driven by 2 pSS patients that developed lymphoma. I do not think the applied statistics is correct since the data points between pSS and pSS with lymphoma highly overlap. Therefore, I suggest that the manuscript should be checked by a statistician.
The conclusion (last sentence) that “miR-155 arises as novel biomarker for patient stratification” is not supported by data, given the high overlap between miR-155 expression between patients with and without lymphoma.
Methods, qPCR: the authors describe that for qPCR results, fold changes relative to control have been calculated. This is a common way of analyzing changes in gene expression e.g. in cell cultures with and without stimulation, where the unstimulated cells can be used as control. Can you please specify what your “control” was in your case (one of the SS patients ?). For comparison of expression levels between two patient populations, it is more common to calculate only delta ct values and not fold changes.
A high focus score is a known risk factor for the development of lymphoma in Sjögren`s Syndrome (e.g. reviewed inV. Manfrè et al., Clin Exp Rheumatol. 2022 Dec;40(12):2211-2224; doi: 10.55563/clinexprheumatol/43z8gu). Can you please discuss your finding with respect to published literature.
Figure 1 lacks novelty. Higher levels of CD20+ and CD138+ cells in tissue sections have been described by Carrillo-Ballestros et al., Clin Exp Med. 2020 Nov;20(4):615-626; doi: 10.1007/s10238-020-00637-0). This study has also included tissues from patients with and without germinal centers, a risk factor for the development of lymphoma. Also, this study has not been discussed in the discussion section.
The same study has also described high levels of BAFF-R in tissues of patients with pSS (+/- GC) compared to controls (Carrillo-Ballestros et al., Clin Exp Med. 2020 Nov;20(4):615-626; doi: 10.1007/s10238-020-00637-0).
Numerous cell types, including different epithelial cells, fibroblasts, lymphocyte populations and others, express miR-155. Based on the data it is not clear which cell type(s) within the labial glands drive the increased expression of miR-155 in patients with pSS. This aspect would be needed to thematically link the individual figures provided in this manuscript.
Minor comments:
Please provide a reference in line 50 after “… chronic inflammation in pSS” and in line 68 after “..process of the disease.”.
Fig. 1C, D: labels of the x-axis has been shifted

Author Response
In this study, the authors aimed to explore the expression of BAFF receptor (BAFF-R) and miR-155 in labial salivary gland tissues of patients with Sjögren`s syndrome (pSS), and the correlation with disease outcome. The strength of the study is that the authors were able to follow up their patients for up to 10 years, enabling them to identify patients who developed lymphoma. With trying to identify biomarkers for lymphomagenesis in patients with pSS, the authors address here an important question of high relevance for patients and clinicians. The novel aspect of the study is the higher expression of miR-155 in tissues of patients with pSS compared to control patients. However, the weakness of this study is its sole descriptive characteristic, the lack of any experimental validation for the role of miR-155 in lymphomagenesis in Sjögren`s Syndrome, and the low number (n=4) of patients who developed lymphoma.
Please find below the specific comments:
Major comments:
In the first sentence of the discussion section the authors mention that “…this study primarily aimed to identify novel potential biomarkers that could help stratifying pSS patients according to their risk of experiencing lymphoproliferative malignancies”. To identify novel biomarkers, one would expect to choose a broader experimental approach rather than pre-selecting two factors (miR-155, BAFF-R).
The conclusion (last sentence) that “miR-155 arises as novel biomarker for patient stratification” is not supported by data, given the high overlap between miR-155 expression between patients with and without lymphoma.
We thank the Reviewer for the comment. We are aware of the limitations of the study, primarily due to its retrospective nature. However, it was conceived as an exploratory study, aiming at detecting potential novel biomarkers of lymphomagenesis in Primary Sjögren's Syndrome. We pre-selected these two factors to deeply investigate the potential role of B-cell lineage in the neoplastic evolution, considering their pivotal role in the pathogenesis of the disease. The novelty of the study consists in studying, for the first time, the possible synergistic interaction between miRNA regulation and the BAFF/BAFF-R axis in Sjogren's disease.
Among the 24 patients with pSS, only 4 patients developed lymphoma. Among these 4 patients, 2 patients exhibit similar fold changes in expression levels for BAFF-R and miR-155 overlap with pSS patients without lymphoma. Therefore, the differences in data (Figs. 2 and 3) are driven by 2 pSS patients that developed lymphoma. I do not think the applied statistics is correct since the data points between pSS and pSS with lymphoma highly overlap. Therefore, I suggest that the manuscript should be checked by a statistician.
We thank the reviewer for the comment. As requested, an extended revision from a statistician was performed to support the presented data findings. In particular, the differential BAFF-R and miR-155 expression between the 3 different patient cohorts depicted in Figure 2 and 3 was determined using one -way ANOVA with correction multiple comparisons by controlling the false discovery rate. The new results were reported in the updated Figures.
Methods, qPCR: the authors describe that for qPCR results, fold changes relative to control have been calculated. This is a common way of analyzing changes in gene expression e.g. in cell cultures with and without stimulation, where the unstimulated cells can be used as control. Can you please specify what your “control” was in your case (one of the SS patients?). For comparison of expression levels between two patient populations, it is more common to calculate only delta ct values and not fold changes.
We want to clarify that the control group used as reference for the qPCR experiments was the sicca syndrome cohort. In particular, for the semi-quantitative determination of the expression of BAFF-R and miR155 and the relative control gene in pSS salivary glands tissue, DCT values were generated after subtraction. From the gene of interest (BAFF-R and miR-155) Ct value of controls. The DDCT values were obtained by normalizing the DCTs with the mean DCt of sicca syndrome group. Relative quantitative values for the expression of miR155 and BAFF-R in salivary gland tissue were calculated as 2^DDCT. This information was added in the method section.
A high focus score is a known risk factor for the development of lymphoma in Sjögren`s Syndrome (e.g. reviewed inV. Manfrè et al., Clin Exp Rheumatol. 2022 Dec;40(12):2211-2224; doi: 10.55563/clinexprheumatol/43z8gu). Can you please discuss your finding with respect to published literature.
We thank the Reviewer for the kind comment. Among the well-known risk factors of lymphomagenesis in Primary Sjögren's Syndrome (pSS), the role of FS and ectopic lymphoid structures as more relevant histopathological stratifiers is well established. In the present study, no clinical and laboratory parameters have been identified to predict the risk of lymphoma. Equally, we found no difference in terms of FS between patients who developed NHL and those who did not. In this scenario, we propose the upregulation of BAFF-R and miR-155 as two potential biomarkers capable of predicting of NHL development in a cohort of pSS patients who did not exhibit the canonical risk factors.
However, we implemented the Discussion section as suggested to highlight this topic.
Figure 1 lacks novelty. Higher levels of CD20+ and CD138+ cells in tissue sections have been described by Carrillo-Ballestros et al., Clin Exp Med. 2020 Nov;20(4):615-626; doi: 10.1007/s10238-020-00637-0). This study has also included tissues from patients with and without germinal centers, a risk factor for the development of lymphoma. Also, this study has not been discussed in the discussion section.The same study has also described high levels of BAFF-R in tissues of patients with pSS (+/- GC) compared to controls (Carrillo-Ballestros et al., Clin Exp Med. 2020 Nov;20(4):615-626; doi: 10.1007/s10238-020-00637-0).
We thank the Reviewer for the kind comment. Indeed, we discussed the results showed by Carrillo-Ballestros et al (reference n° 49), which propose the BAFF system as a major contributor to focal lymphocytic infiltration as well as to the establishment of ectopic GC-like structures in labial salivary glands.
Numerous cell types, including different epithelial cells, fibroblasts, lymphocyte populations and others, express miR-155. Based on the data it is not clear which cell type(s) within the labial glands drive the increased expression of miR-155 in patients with pSS. This aspect would be needed to thematically link the individual figures provided in this manuscript.
We thank the Reviewer for the appropriate comment. We absolutely agree with the point arisen by the Reviewer. Indeed, in the discussion section, we stated that the miR-155 and BAFF-R expression in glandular tissue by individual cytotypes is unknown, given that immunofluorescence was not performed as confirmation. (lines 330-336).
This is definitely a caveat of the study. However, we can postulate that B-cells might be the primary source of increased expression as IHC analysis of glandular tissue showed a higher predominance of B lymphocytes and plasmacells in pSS patients compared to SS, which is strongly consistent with the literature. (lines 330-336).
Thus, as suggested, we modified the Figures to make more clear it.
Minor comments:
Please provide a reference in line 50 after “… chronic inflammation in pSS” and in line 68 after “..process of the disease.”.
We thank the Reviewer for the kind comment which improves the quality of the manuscript. As suggested, we provided two new references.
Fig. 1C, D: labels of the x-axis has been shifted
We thank the Reviewer for the kind comment. As suggested, we improved the quality of Figure 1.
Round 2
Reviewer 4 Report
Comments and Suggestions for Authors
The differences in miR-155 are still driven by 2 data points in the lymphoma group. It is not possible based on the few data points to suggest miR-155 and BAFF-R as biomarkers for lymphoma.
I still believe that some aspects (everything in Fig. 1, and BAFF-R expression in salivary gland tissues) lacks novelty. In total, there is one Figure with novel results (miR-155 in salivary gland tissues) and this is weak because of the overlap between sample groups and the few data points.
Author Response
The differences in miR-155 are still driven by 2 data points in the lymphoma group. It is not possible based on the few data points to suggest miR-155 and BAFF-R as biomarkers for lymphoma.
I still believe that some aspects (everything in Fig. 1, and BAFF-R expression in salivary gland tissues) lacks novelty. In total, there is one Figure with novel results (miR-155 in salivary gland tissues) and this is weak because of the overlap between sample groups and the few data points.
We thank the Reviewer for the comment. We are aware of the limitations of the study, primarily due to its retrospective nature. However, it was conceived as an exploratory study, aiming at detecting potential novel biomarkers of lymphomagenesis in Primary Sjögren's Syndrome. We pre-selected these two factors to deeply investigate the potential role of B-cell lineage in the neoplastic evolution, considering their pivotal role in the pathogenesis of the disease. The novelty of the study consists in studying, for the first time, the possible synergistic interaction between miRNA regulation and the BAFF/BAFF-R axis in Sjogren's disease.
To point out this concept, we added a statement in the Discussion section, as suggested by the academic Editor.
However, it should be noted that among patients with Primary Sjögren's Syndrome (pSS) who have not developed lymphoma, the three presenting the highest values of miR-155 tissue expression, comparable to those of patients who had neoplastic evolution, showed a significant B-cellular activation, with an increased hypergammaglobulinemia, high titres of anti-SSA and anti-SSB antibodies and Focus Score ≥ 5. Therefore, this sets them up as patients at high risk of developing lymphoma over the course of the disease. It is useful to remember here that all pSS patients not developing NHL had a follow-up of 10 years.
I attached you the table with demographic, immunological and histopathological characteristics of these three patients.
